# Prediction of postoperative patient deterioration and unanticipated intensive care unit admission using perioperative factors

Eveline H. J. Mestrom[1]☯*, Tom H. G. F. Bakkes[2]☯, Nassim Ourahou[1], Hendrikus H. M. Korsten[1,2], Paulo de Andrade Serra[3], Leon J. Montenij[1], Massimo Mischi[2], Simona Turco[2], R. Arthur Bouwman[1,2]

1 Anesthesiology Department, Catharina Hospital Eindhoven, Eindhoven, The Netherlands, 2 Signal Processing Department, Eindhoven University of Technology, Eindhoven, The Netherlands, 3 Mathematics Department, VU Amsterdam, Amsterdam, The Netherlands

☯ These authors contributed equally to this work.
* eveline.mestrom@catharinaziekenhuis.nl

**Data Availability Statement:** The relevant data is uploaded to Dryad Data Repository at the following DOI: https://doi.org/10.5061/dryad.66t1g1k6g.

## Abstract

### Background and objectives

Currently, no evidence-based criteria exist for decision making in the post anesthesia care unit (PACU). This could be valuable for the allocation of postoperative patients to the appropriate level of care and beneficial for patient outcomes such as unanticipated intensive care unit (ICU) admissions. The aim is to assess whether the inclusion of intra- and postoperative factors improves the prediction of postoperative patient deterioration and unanticipated ICU admissions.

### Methods

A retrospective observational cohort study was performed between January 2013 and December 2017 in a tertiary Dutch hospital. All patients undergoing surgery in the study period were selected. Cardiothoracic surgeries, obstetric surgeries, catheterization lab procedures, electroconvulsive therapy, day care procedures, intravenous line interventions and patients under the age of 18 years were excluded. The primary outcome was unanticipated ICU admission.

### Results

An unanticipated ICU admission complicated the recovery of 223 (0.9%) patients. These patients had higher hospital mortality rates (13.9% versus 0.2%, p<0.001). Multivariable analysis resulted in predictors of unanticipated ICU admissions consisting of age, body mass index, general anesthesia in combination with epidural anesthesia, preoperative score, diabetes, administration of vasopressors, erythrocytes, duration of surgery and post anesthesia care unit stay, and vital parameters such as heart rate and oxygen saturation.

**Funding:** The author(s) received no specific funding for this work.

**Competing interests:** The authors have declared that no competing interests exist.

The receiver operating characteristic curve of this model resulted in an area under the curve of 0.86 (95% CI 0.83–0.88).

## Conclusions

The prediction of unanticipated ICU admissions from electronic medical record data improved when the intra- and early postoperative factors were combined with preoperative patient factors. This emphasizes the need for clinical decision support tools in post anesthesia care units with regard to postoperative patient allocation.

## Introduction

Currently, no evidence based criteria exist for decision making with regard to postoperative patient allocation in the post anesthesia care unit (PACU). An unanticipated intensive care unit (ICU) admission is the result of a serious complication in postoperative patients. Despite improvements in anesthesia and postoperative care, 14–17% of patients undergoing surgery suffer from serious postoperative complications [1–3]. Approximately 1% of these patients are transferred to the intensive care unit (ICU) due to serious deterioration [4–6]. In addition, unanticipated critical care admissions are associated with higher mortality rates than planned critical care admissions [6]. In addition to the impact on patient health and outcomes, there are negative consequences, such as less efficient allocation and management of limited ICU resources.

In current practice, the postoperative patient in the post anesthesia care unit (PACU) depends on the expertise of nurses and finally the anesthesiologist who decides if the patient is sufficiently clinically stable for transfer to the ward. Clinical experience and knowledge are primarily used to support clinical decision making but these are subject to multiple factors such as fatigue, cognitive overload, busy schedules and capacity in the hospital. Current discharge criteria, such as Aldrete's scoring system, do not integrate factors from all perioperative stages to support clinical decision making [7]. Although an increasing amount of patient data is stored in the electronic medical record (EMR), these data are not systematically used and included for systematic assessments in the PACU. With the implementation of advanced EMRs, the readily available data from all perioperative stages in the EMR could improve the development of clinical decision support tools in the PACU by assigning patients an automatically calculated risk score for unanticipated ICU admission.

A previous study found that including pre- and postoperative variables improves the prediction of postoperative deterioration [5]. While this already underlines the importance of using the available EMR data, the study design was of a prospective nature and included prospectively collected observations. Evidence suggests that favorable outcomes in postoperative patients could be achieved by pre-emptive cardiorespiratory interventions, such as (non)invasive ventilation and inotropic or vasopressor support, which require admission to a higher acuity department [8, 9]. However, providing these interventions to the majority of postoperative patients is not realistic, as high care units and human resources are limited. Therefore, the identification of predisposing events for deterioration in the operating theatre and PACU might be crucial to improve patient safety.

The aim of this study was to assess the value of routinely collected perioperative data for the prediction of postoperative deterioration in terms of unanticipated ICU admission. We

hypothesized that the use of meaningfully selected data could provide a basis for data-driven decision support tools in post anesthesia care units.

## Methods

A retrospective cohort study was conducted at Catharina Hospital, a tertiary 696-bed training hospital in Eindhoven, The Netherlands. The study was approved by the Medical Research Ethics Committees United (MEC-U local number W18.071), Nieuwegein, The Netherlands The requirement for written informed consent was waived. This manuscript adheres to the applicable TRIPOD guidelines.

The study hospital performs approximately 7400 surgical procedures admits 3000 patients to ICU annualy. The majority of patients in the ICU are admitted following cardiothoracic surgery and are discharged within 48 hours. Furthermore, the ICU population is characterized by postoperative major abdominal surgery, medical and drug overdose but very few patients following neurotrauma or neurosurgery, or transplant patients. In the preoperative outpatient clinic, it is determined by the attending anesthesiologist whether ICU admission or surgical ward admission is anticipated after surgery. This preoperative planning is mostly based on the American Society of Anesthesiology (ASA) score and a list of surgeries that require postoperative ICU admission, such as cardiothoracic surgeries, per protocol. In case the decision was made during the screening to transfer the patient to the surgical ward after surgery, the patient would recover in the PACU until discharge to the ward when predefined discharge criteria such as Aldrete's scoring system were met. When a patient is not recovering according to expectations, the anesthesiologist is consulted and decides on the future care requirements for the patient. Options are discharge to the ward, discharge to the ward after prolonged stay in the PACU, or admission to the ICU.

### Data collection

Unanticipated ICU admission was classified as the creation of an ICU record more than two hours after the last recorded heart rate in the PACU, meaning that the patient was discharged to the ward after the PACU stay before unanticipated ICU admission occurred (Fig 1). In case of planned ICU admission, the patient is transferred directly from the operating theatre to the ICU. The authors manually reviewed all 285 cases identified by the HR rule and excluded any instances where an unanticipated ICU admission did not occur, or if appropriate moved them to the control group"

Patient data were collected from the electronic medical record (EMR) for every surgical procedure from January 2013 until December 2017. This study period and study size were

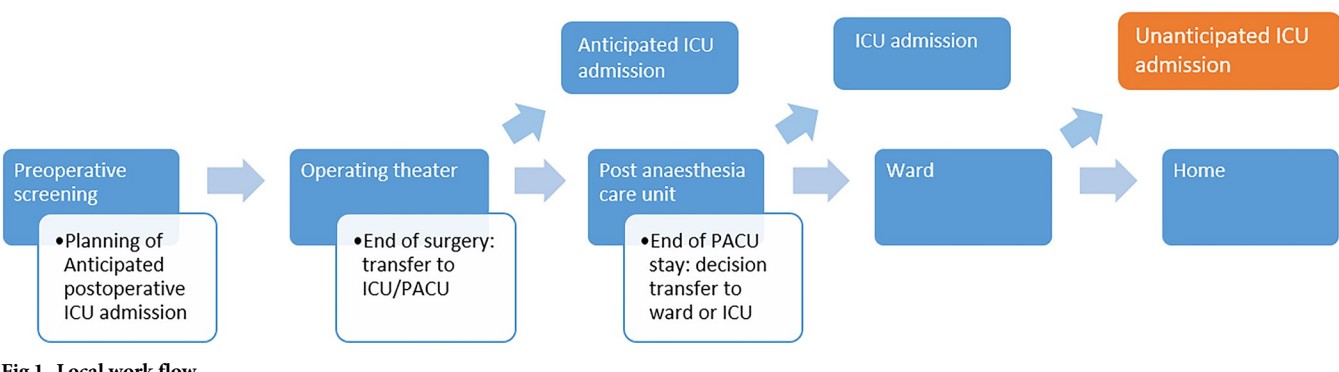

**Fig 1. Local work flow.**

chosen to obtain as many unanticipated ICU admissions as possible without changes in software in either the EMR or operating theatre. Only data with regard to the first main surgery per patient were included to avoid the influence of previous surgical procedures. Cardiothoracic surgery, obstetric surgery, catheterization lab procedures, electroconvulsive therapy, day care procedures, intravenous line interventions and patients under the age of 18 years were excluded since they are different categories in terms of postoperative care and logistics. Vascular surgery was included. Data available in the EMR from preoperative screening, intra- and postoperative vital signals, medication, blood products, events registered in the operating theatre, date and times of intervention of emergency teams and transfers to the ICU were collected. A detailed overview can be found in S1 Table. Data consisted of both categorical and continuous variables. Intraoperative data were collected in the EMR via AnStat software version 2.0.6, Carepoint B.V., which automatically records the intraoperative variables in the EMR and where remarks by perioperative staff were manually added.

## Statistical analysis

The cohort was divided into a group consisting of postoperative patients who experienced unanticipated ICU admission during their stay in the hospital and a group consisting of postoperative patients without ICU admission during their hospital stay.

Data analysis was performed using MATLAB® (MathWorks Inc., Natick, MA). For comparison of groups, the chi-square test and Fisher's exact test were used for categorical variables. The Mann-Whitney U test was used for continuous variables since data were not normally distributed. All continuous variables were plotted against the logit predictions and visually inspected to determine linearity. The level of significance was set as $p$-value $<0.05$. To control for confounding factors in this study, a multivariable logistic regression was chosen. First, univariate analysis was performed for all variables in the collected data to assess the association with unanticipated ICU admission. Second, Benjamini-Hochberg correction was applied to minimize the multiple statistical testing problem, allowing a 5% false discovery rate. Based on the univariate analysis and Benjamini-Hochberg correction, the 33 significant variables were considered for inclusion as potential confounders in multivariable logistic regression. Multivariable models were built using penalized logistic regression with the L1 loss. Multivariable model building in Statistics and Machine Learning Toolbox in MATLAB was performed using the ' lassoglm' function. During the training, the L1 scaler was fitted using 3 fold cross-validation, with a grid search over a 100 scalar values. The ratio between the maximum and minimum of the grid search values was 1e-4. Models were cross-validated using bootstrapping repeated a 100 times. During the bootstrapping, the dataset would be resampled with replacement to form the training dataset. The remaining out-of-bag samples were used as the test set. ROC curves were examined for comparison between the optimal model using pre-, intra- and early postoperative data and a model containing only preoperative variables that are readily available in the EMR. Missing data were not replaced or imputed. Patients with missing variables were excluded from the multivariable analysis. Bias from missing data was expected to be low, as most of the data were registered in the EMR automatically.

## Results

Computer-guided identification yielded 25,292 controls and 285 cases. After manual checking in the EMR, the final group consisted of 25,296 controls and 223 cases (Fig 2). Due to missing variables, a total of 21526 controls and 179 cases were included for analysis.

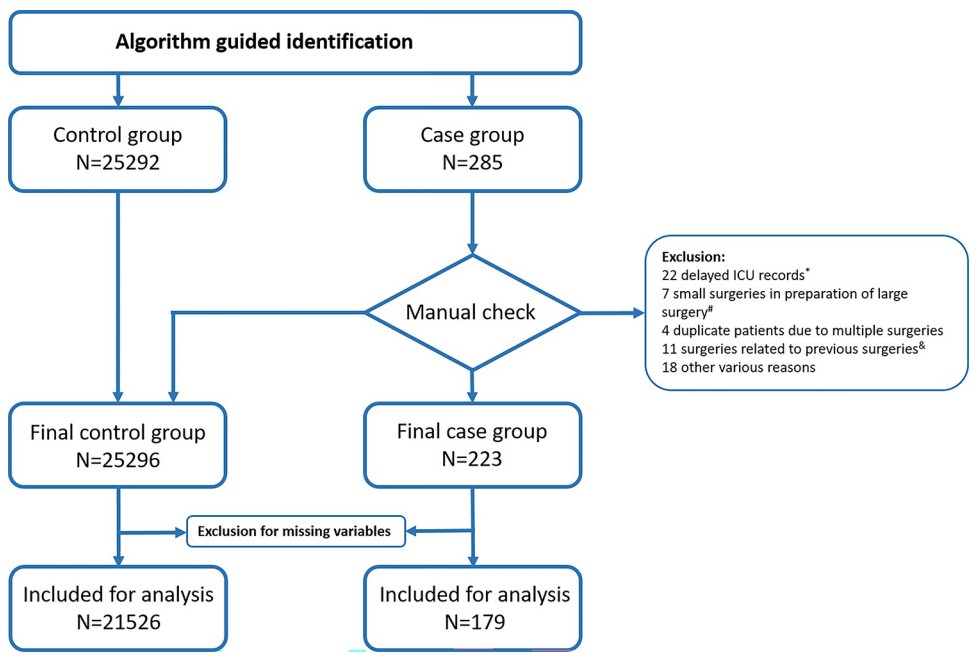

**Fig 2. Flowchart of selection procedure.**

## Patient characteristics

All clinically relevant variables were compared between cases and controls. The most statistically significant findings are presented in S2 Table. The median time between PACU discharge and unanticipated ICU admission was 2.68 days (IQR 4.61 days). A histogram on these time spans is presented in S1 Fig. In-hospital mortality was higher among the case group (13.9% vs. 0.2%, p<0.001). Within the case group, 27 of the 31 (87.0%) patients died in ICU. With regard to comorbidities, diabetes, hypertension, cerebrovascular accident and thromboembolic events were significantly more associated with unanticipated ICU admission. Additionally, antiplatelet drugs (14.3% vs. 3.3%, p<0.001) and vitamin K antagonists (16.6% vs. 4.0%, p<0.001) were prescribed significantly more often in the case group. Patients who required an unanticipated ICU admission were significantly older, underwent longer surgeries and stayed longer in the PACU, had higher ASA Physical Status Classification System scores and required more hemodynamic support during surgery.

Following surgery, the cases required more frequent review by the anesthesiologist (13.0% vs. 5.1%, p<0.001). Cases experienced more abnormalities in vital parameters, of which oxygen saturation below 85% was the most notable (27.4% vs. 13.2%, p<0.001).

Univariate analysis was performed for all clinically relevant variables of interest. These results can be found in S3 Table. Variables with few to no numbers were exempt from univariate analysis.

## Multivariable analysis

Binomial logistic regression utilizing penalized regression yielded the multivariable model with the strongest predictors and an optimal AUC-ROC value of 0.85 (95% CI 0.82–0.88). The most important variables in the prediction are listed in Table 1. The list is based on the odds ratios of the predictor during bootstrapping. A predictor is included in this list if during the bootstrapping the 95% confidence interval of the odds ratio did not contain an odds ratio of 1.

**Table 1. Most important predictors resulting from penalized regression.**

| | | OR | CI (95%) |
|---|---|---|---|
| **Preoperative period** | | | |
| Age, years | | 1.18 | 1.01–1.34 |
| Diabetes mellitus | | 1.07 | 1.00–1.17 |
| ASA[a] score 1 | | 0.86 | 0.73–1.00 |
| ASA[a] score 3 | | 1.23 | 1.12–1.35 |
| Anesthesia technique | | | |
| | General | 1.00 (reference) | |
| | General and epidural | 1.17 | 1.09–1.26 |
| | Spinal | 0.97 | 0.87–1.00 |
| | Other | 0.90 | 0.79–0.99 |
| **Surgery period** | | | |
| Administration of vasopressors | | 1.18 | 1.08–1.28 |
| Transfusion of red blood cells | | 1.05 | 1.00–1.09 |
| Time in operating theatre | | 1.20 | 1.01–1.34 |
| Surgery group (General surgery) | | 1.57 | 1.35–1.83 |
| **PACU[b] period** | | | |
| Heart rate >100 bpm | | 1.10 | 1.00–1.24 |
| Minimum heart rate | | 1.20 | 1.06–1.33 |
| Oxygen saturation <85% | | 1.07 | 1.00–1.17 |
| Time in PACU | | 1.37 | 1.25–1.48 |

[a]ASA: American Society Anesthesiologists
[b]PACU: Post Anesthesia Care Unit.

This list contains preoperative, intraoperative and postoperative variables. The full list containing all predictors can be found in S4 Table. Abnormalities in vital parameters during and after surgery were revealed to be strong predictive variables. The multivariable analysis showed that the anesthesiologists' review had a stronger correlation with the ASA Physical Status Classification System score and duration of stay in the PACU than with unanticipated ICU admission and was therefore regarded as a confounder.

The AUC-ROC of the best model is shown in Fig 3. The AUC of 0.85, including pre-, intra- and postoperative data, was higher than that including only preoperative data or pre- and intraoperative data, as shown in S2 and S3 Figs. The model has a calculated accuracy of 0.98, precision of 0.14 and recall of 0.17 with a resulting F1 score of 0.15. The AUPRC was 0.09 (95% CI 0.05–0.14) and is shown in Fig 4.

Regarding characterization of the unanticipated ICU admissions, cardiovascular organ dysfunction was most prevalent (49.3%), followed by hematological complications (22.9%) and respiratory insufficiency (19.7%). A vast majority of cardiovascular dysfunction and respiratory insufficiency events arose from infectious causes, and in 70% of the cases, antibiotics were prescribed. Vasopressor and inotrope support were required in 39% and 9.4% of the cases, respectively. Mechanical ventilation was mostly required in patients who suffered from cardiovascular dysfunction due to abdominal sepsis with subsequent respiratory insufficiency. The complete results are shown in S5 and S6 Tables.

## Discussion

This study showed that the inclusion of perioperative data improved the predictive value of postoperative unanticipated ICU admission. The main predictors might not be surprising but

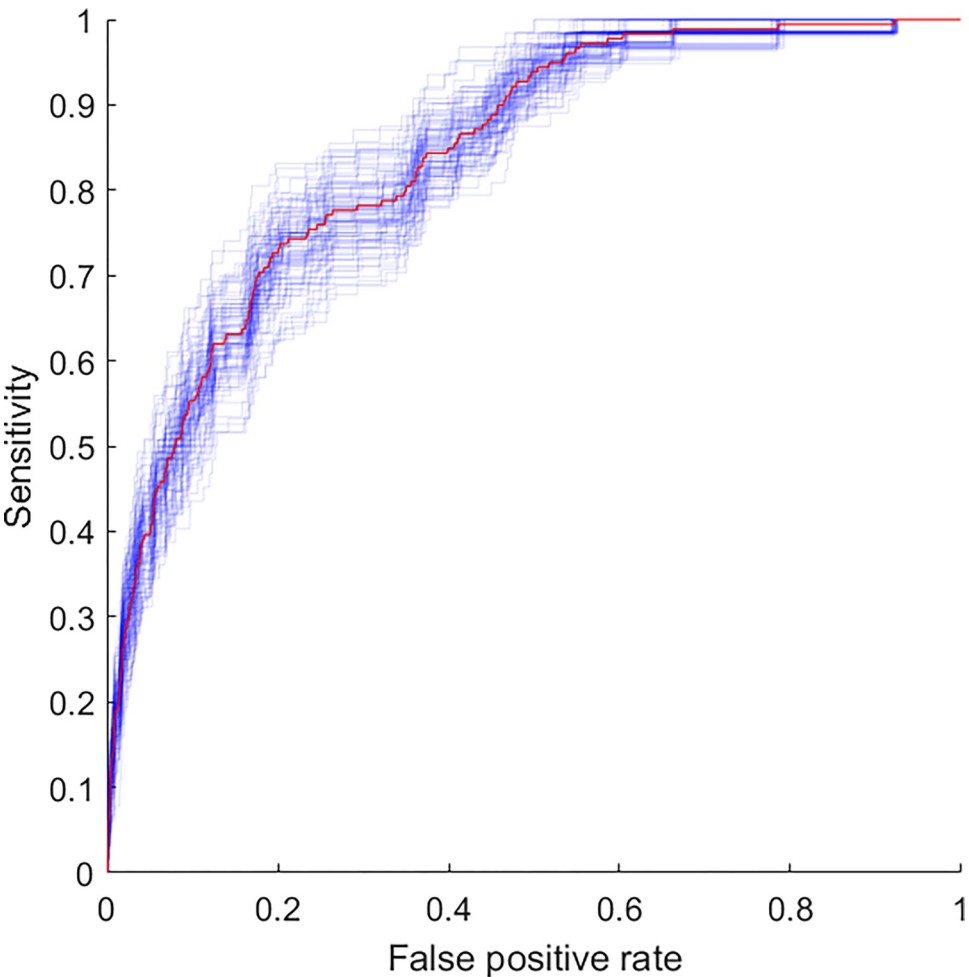

**Fig 3. ROC curve including pre-, intra- and postoperative data.**

were readily available from the EMR: ASA Physical Status Classification System score, duration of surgery, general anesthesia combined with epidural analgesia, transfusion of erythrocytes, heart rate >100 bpm and postoperative oxygen saturation <85%. These study results emphasize the importance to incorporate these informative data in future clinical decision support tools in PACU.

The approach in this research is comparable to the study by Petersen Tym, who reported similar findings [5]. Although their prospective study was better designed to avoid missing data, the retrospective design better reflects what kind of readily available information a decision support tool would find in the EMR. The method of including intraoperative data improved prediction in cardiothoracic patients undergoing lung resection surgery, although different intraoperative variables were included [10]. A recent systematic review consistently found a high average intraoperative heart rate, low mean arterial pressures, increased blood loss and operative duration as independent risk factors in multivariable analysis throughout the included studies [11, 12]. These studies demonstrate that even in different populations and different variables, intraoperative data are of value for the prediction of postoperative adverse events. An interesting methodological approach using intraoperative data was the comparison of deep neural network prediction versus conservative logistic regression models by Lee et al.

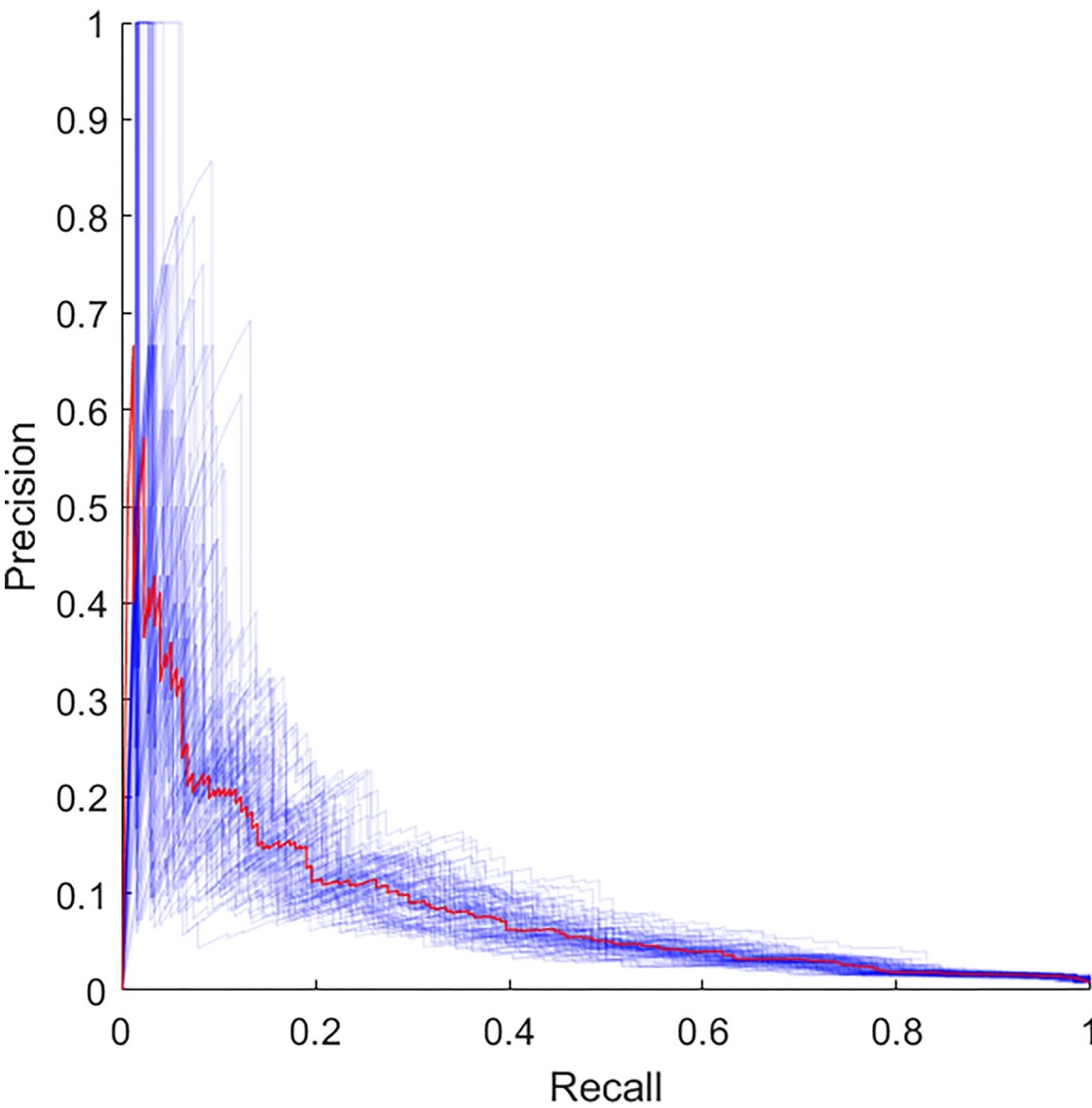

**Fig 4. AUPRC including pre-, intra- and postoperative data.**

[13]. Their results using deep neural networks showed slightly better AUCs than logistic regression models and a reduced number of false positives. However, the primary outcome in the study by Lee et al. was in-hospital mortality, which challenges comparison of performance of the models with our study.

The unanticipated ICU admission rate was 0.9% in this study, which is consistent with findings in the literature across different countries in Europe and Australia [5, 6]. Mortality was higher among the cases (13.9%), which is in line with the results in different countries in Europe [6].

The findings in this study provide valuable insights into postoperative deterioration resulting in unanticipated ICU admission. Even in early postoperative situations, the data in this study established reasonable predictive value from a PACU perspective, which is of interest to

anesthesiologist. The results are in line with expectations from a clinical point of view, suggesting that algorithms are capable of recognizing or even predicting deterioration. The knowledge from this study is important to ground how useful it can be to support decisions based on data. The results of a predictive model can help prioritize patient care with the same approach as an early warning score. At the end of the PACU stay the model can provide a prediction on how likely the patient is to deteriorate and highlight high-risk patients. For these patients, it can then be decided to send them to the ICU instead of the ward or to provide them with a higher level of monitoring (e.g. more frequent spot checks).

There were several limitations in the present study. First, assumptions were made for an automatic screening algorithm to identify unanticipated ICU admissions, as they were not clearly marked in the EMR. This exposes the limitations of current data structures that have not yet been designed for EMR data-based algorithms. The missing structured information on unanticipated ICU admissions was overcome by manual screening in the case group but remains undesirable for future purposes. In addition, comorbidities were poorly registered in the EMR, especially in patients undergoing emergency surgery. This can be explained by the limited time available to perform or document a complete preoperative screening for emergency patients. Second, given how unusual unanticipated ICU admissions are, this study was conducted on a small number of heterogeneous cases. The small number of cases (N = 223) compared to controls (N = 25,296) biases the results towards negative predictions; unfortunately, it is challenging to correct for this bias due to the high number of variables (N = 48) using upsampling or weighted logistic regression. Third, this study was performed in a single center and without a validation cohort. Local procedures and intervention thresholds may vary and therefore may not be applicable in other centers. For example, The Netherlands has 6.4 ICU beds per 100,000 population, compared to 28 per 100,000 in the United States of America [14]. And on the other end of the spectrum, an estimated five billion individuals in low-resource countries are subject to delays and shortages in perioperative care [15]. These numbers could influence local differences, such as prophylactic or pre-emptive ICU admissions, which was found to be an important factor for the use of ICU admission as an outcome measure [16]. Fourth, this study was not designed to demonstrate improved outcomes if better allocation of postoperative patients would be chosen, but this remains an important issue as described by other researchers [17]. Fifth, penalized logistic regression appeared to be superior to conventional multivariable analysis, using a p-value <0.05 as a criterion for variable selection in the univariate analysis. The drawback of this method is that variables that could improve performance in combination with other variables in a multivariable model might be excluded. In the end, the results were still good but could perhaps even be better had the limitations not been present.

This study showed the advantage of using perioperative data. The next step is the development of a digital tool to automatically assign risk scores for deterioration such as unanticipated ICU admission. This digital tool could automatically calculate a low, intermediate, or high risk of unanticipated ICU admission and provide decision support to the anesthesiologist in PACU. Future medical research could focus on more advanced probabilistic learning methods [13]. For instance, Bayesian networks permit leveraging medical expert knowledge by permitting selection of relevant predictors and design of the model structure, which enables the definition of causal relations between predictors [18]. Updating models based on new evidence or computer-guided pattern recognition in newly available data is promising, as these feature-rich models appear to have greater accuracy than conventional methods and less limited by granular or missing data [19, 20]. These techniques could be used to develop real-time decision support tools that can be implemented in daily medical practice.

## Conclusion

The prediction of unanticipated ICU admissions from readily available EMR data improved when the intra- and early postoperative factors were combined with preoperative patient factors. This emphasizes the need for clinical decision support tools in post anesthetic care units with regard to postoperative patient allocation.

## Supporting information

**S1 Fig. Histogram of time spans.**
(TIF)

**S2 Fig. AUC including pre-, intra- and postoperative data.**
(TIF)

**S3 Fig. AUC including pre- and intraoperative data.**
(TIF)

**S1 Table. Collected variables.**
(DOCX)

**S2 Table. Baseline characteristics and outcomes.**
(DOCX)

**S3 Table. Results of univariate analysis.**
(DOCX)

**S4 Table. Predictors after bootstrapping.**
(DOCX)

**S5 Table. Types of organ dysfunction as underlying reason for unanticipated ICU admission.**
(DOCX)

**S6 Table. Interventions in ICU during unanticipated ICU admission.** Each category is further subdivided into involved organ systems as underlying reason for unanticipated ICU admission.
(DOCX)

## Author Contributions

**Conceptualization:** Eveline H. J. Mestrom, Tom H. G. F. Bakkes, Nassim Ourahou, Hendrikus H. M. Korsten, Paulo de Andrade Serra, Massimo Mischi, Simona Turco, R. Arthur Bouwman.

**Data curation:** Tom H. G. F. Bakkes, Nassim Ourahou.

**Formal analysis:** Eveline H. J. Mestrom, Tom H. G. F. Bakkes, Paulo de Andrade Serra.

**Investigation:** Eveline H. J. Mestrom, Tom H. G. F. Bakkes, Nassim Ourahou.

**Methodology:** Eveline H. J. Mestrom, Tom H. G. F. Bakkes, Nassim Ourahou, Hendrikus H. M. Korsten, Paulo de Andrade Serra, Leon J. Montenij, Massimo Mischi, Simona Turco, R. Arthur Bouwman.

**Resources:** Eveline H. J. Mestrom, Nassim Ourahou.

**Supervision:** Hendrikus H. M. Korsten, Paulo de Andrade Serra, Leon J. Montenij, Massimo Mischi, Simona Turco, R. Arthur Bouwman.

**Validation:** Tom H. G. F. Bakkes.

**Visualization:** Tom H. G. F. Bakkes.

**Writing – original draft:** Eveline H. J. Mestrom.

**Writing – review & editing:** Tom H. G. F. Bakkes, Nassim Ourahou, Hendrikus H. M. Korsten, Paulo de Andrade Serra, Leon J. Montenij, Massimo Mischi, Simona Turco, R. Arthur Bouwman.

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
