## [Decision Letter · Decision Letter 0]

4 Jul 2022

PONE-D-22-01611Prediction of postoperative patient deterioration and unanticipated intensive care unit admission using perioperative factorsPLOS ONE

Dear Dr. Mestrom,

Thank you for submitting your manuscript to PLOS ONE. After careful consideration, we feel that it has merit but does not fully meet PLOS ONE’s publication criteria as it currently stands. Therefore, we invite you to submit a revised version of the manuscript that addresses the points raised during the review process.

We look forward to receiving your revised manuscript.

Kind regards,

Sandro Pasquali, M.D., Ph.D.

Academic Editor

PLOS ONE

https://journals.plos.org/plosone/s/file?id=ba62/PLOSOne_formatting_sample_title_authors_affiliations.pdf".

Additional Editor Comments:

Carefully address reviewers' comments with special regard to methodological and biostatical comments.

Reviewers' comments:

Reviewer's Responses to Questions

**Comments to the Author**

1. Is the manuscript technically sound, and do the data support the conclusions?

Reviewer #1: No

Reviewer #2: Partly

Reviewer #3: Yes

Reviewer #4: Partly

2. Has the statistical analysis been performed appropriately and rigorously? 

Reviewer #1: No

Reviewer #2: I Don't Know

Reviewer #3: Yes

Reviewer #4: No

3. Have the authors made all data underlying the findings in their manuscript fully available?

Reviewer #1: Yes

Reviewer #2: Yes

Reviewer #3: Yes

Reviewer #4: No

4. Is the manuscript presented in an intelligible fashion and written in standard English?

Reviewer #1: Yes

Reviewer #2: Yes

Reviewer #3: No

Reviewer #4: Yes

5. Review Comments to the Author

Reviewer #1: Overall comments:

This study was based in a large tertiary hospital in The Netherlands and sought to develop a prediction model utilising pre, intra- and post-operative factors to predict an unanticipated ICU admission. An unanticipated ICU admission was defined as the patient being discharged from PACU to the ward, but then needing ICU admission. Over 25,000 patients were included, with 223 patients meeting the definition of a case. Using factors from the pre, intra- and post-operative the AUROC was highest, compared to only included pre-operative or pre- and intra-operative factors.

I have structured my reviews below with section and line specific comments. But firstly some over-arching comments.

I would reccommend major revisions for this paper. In the current form it is not appropriate for publication. However, the paper is well written, methodology relatively sound and conclusions not far-reaching. If the below suggestions can be considered and if possible addressed I would be very happy to re-review this paper and consider it for acceptance.

There are 4 main problems that must be addressed:

#1 - is this paper seeking to create a prediction tool or compare the type of data (pre/intra/post-operative data) included in the tool to determine which is best? I would suggest including all data that will be known at the time of making the decision. If the authors disagree that is fine, but it needs to be clear in all sections of the paper, the abstract and methods state you are comparing how different data influences the model, but the introduction does not. Further, I am not clear on the benefit of assessing what data types to include, why not include all available data and build the best possible model? It sounds like all data is available in the EMR anyway?

#2 - Data is included in the model that would not be available when you wish to use the model. Looking through table 1 the 4 PACU period variables would not be known at the time of using the model (pre PACU discharge). The authors need to redo their analysis with only variables known prior to PACU discharge

#3 - Is there an upper limit for how long post PACU discharge an ICU admission was defined as a case? If you have included ICU admissions >2-3 days after being on the ward I worry if you case cohort is not homogenous. The reasons and contributing factors to ICU admission immediately after PACU discharge and after being on the ward for >2-3 days are very different. I would advocate for you to report median/mean + IQR/SD (or better yet a histogram) for the time interval from PACU discharge to ICU admission to help readers understand and limit your cohort to not include admissions after >2-3 days post PACU discharge/

#4 - You need to explain how this model can be used practically. Would it be implemented into an EMR? Or can you choose 4-5 of the variables, assign scores for each variable that contribute to a global score that corresponds with a X% change of needing to go to ICU unexpectedly. Without clearly providing a path for implementation a model is of little benefit to the medical world and most importantly, patients. Practically speaking if you could create a tool for readers that uses your data to assign a % chance of having an unplanned ICU admission that would be helpful. The tools I use in my day to day practice as easy to calculate, contain few variables and give me a low / medium / high risk fo X outcome.

Abstract:

L41 Post anaesthesia care unit stay - add "length of" to the start. Currently it sounds like a binary variable.

Introduction:

- Overall a well written introduction. Authors often struggle to explain the "why" for prediction tools like this, but you have done this well. One overall comment is the final section of the introduction leads me to believe you are aiming to create a prediction tool, whereas your abstract makes it sounds like you are assessing the benefit of including intra- and postoperation factors to a tool that already has pre-operative factors (particularly in your abstract conclusion). As a reader I am left wondering - are these authors creating a new tool? Or are they trying to alter an existing tool to make it better? You need to clarify this so readers don't get confused and have a clear path of reading to follow.

L59 The authors reference critical care admissions -> I would stick with referring to ICU admissions for consistency.

L66 This line needs rewording "Most clinical experience..." -> I would sugggest changing it to "Clinical experience and knowledge are primarily used to support clinical decision making..."

L68 would suggest changing brain overload to cognitive overload

Methods:

- Overall a clear methods section. A few things need to be clarified, primarily the method of manual case review. Further, you need to clarify if there is an upper bound for how long post PACU discharge an ICU admission is considered a case. If you are including patients that go to ICU >2-3 days post PACU discharge this may impact your models ability to predict these admissions. As one can imagine intra and post-operative factors are unlikely to predict ICU admission 2-3 days after the surgery.

L94 Unsure if this is specific to regulations in The Netherlands (if so please disregard my suggestion), but including the date and chairperson of the committee that approved your project is probably uneccesary. The ethics committee and reference number is sufficient.

L96 STROBE guidelines are not appropriate here. You need to use the TRIPOD guidelines. (https://www.equator-network.org/reporting-guidelines/tripod-statement/)

L97 "In this hospital" - I suggest rewording this. Perhaps "The study hospital performs approximately 7400 surgical procedures and admits 3000 patients to ICU annually"

L100 Suggest changing "large abdominal surgery" to "major abdominal surgery"

L101 Could you explain the psychiatric patients, are these post overdose? If so I would suggest reporting as "drug overdose" instead. When I hear "psychiatric patients" I immediately think they are transfers from a psychiatric unit.

L115 I would suggest adding a line to state you did a manual review of unaticipated ICU admission cases identified by your HR rule. You describe this process in Figure 1, and mention it in the results, but this is not 100% clear. Something like "The authors manually reviewed all 285 cases identified by the HR rule and excluded any instances where an unanticipated ICU admission did not occur, or if appropriate moved them to the control group"

Results:

- From a clinic perspective a performance model only really works if all variables are known, and thus can be included in said model at the time of making a decision. In this case that decision would be made about an hour after finishing the operation when the patient has stabilised in the PACU. Looking at your Table 1 model - the PACU period variables may not be known then. I assume the period for collecting these variables were from entering PACU to leaving it. Unfortunately, if I want to have a model assist me with my decision to transfer a patient to ICU post-operatively I will not know the total time in PACU for example, I will only know how long they have spent so far. Based on this I think including pre-operative and intra-operative variables only is fair. Otherwise your model will be including information that the person using the model will not have access to at the time of making their decision.

Figure 1

- Please tidy up the arrows so they make contact with boxes. Suggest making this in Powerpoint (much easier for arrow aligning). Also make it a consistent colour (all blue border or all black border).

In the exlusion group a few queries:

- What do you mean by "delayed ICU records"

- Why did you exclude "small surgeries"? Did you then include their second surgery as the index operation?

- "Surgeries related to previous surgeries" - what does this mean? Was this their second surgery that you were meant to exclude?

Table 1

- For continuous variable stipulate your units increase. I assume it is +1 year for age and +1kg/m^2 for BMI etc but this needs to be clarified, particularly for time in operating theatre etc.

- ASA score is not continuous, it is a categorical variable. I would suggest coding it as such.

- For the continuous variables did your analysis fulfill the linearity assumption? Claifying this is important. If these did not fulfill that assumption they need to be altered to categorical variables, or adjusted accordingly (log adjust etc)

L216 How did they determine the complications arose from infectious casuses?

Discussion:

- Overall well written and clear. I do challenge the authors to explain where, how and why this model will not be used. Many papers produce prediction models of varying quality, but very few explain how they will use this data. As a reader I can't use this information to implement in my own centre, are you now using it at your own hospital? Is it integrated in your EMR now? It is important to not only produce the prediction model, but to also implement it. This paper provides an example of what to be careful of (https://www.bmj.com/content/369/bmj.m1328). Published Jan 2021 found 232 prediction models centred around COVID, but only 2 models validated in multilpe cohorts that may be appropriate for implementation.

Reviewer #2: The investigators undertook a retrospective review of electronic patient information in a hypothesis-generating attempt to better understand factors that influence admission to the intensive care unit after an operation. They are particularly interested in what they define as an unplanned admission.

I congratulate the team for examining factors that may help to inform their clinical team at their institution. This seems like the beginning of a quality improvement project (a needs assessment). As I understand this, the investigators are trying to identify factors that may be missed from clinicians who plan admissions to the ICU. Or factors that arise during the operation. Other reports in the literature do not focus on unplanned admissions because this outcome is not all that well defined and other more meaningful outcomes (morbidity and mortality) are typically studied in this context. However, I could see how the authors' institution would be interested in the unplanned admissions for planning and cost purposes. Ultimately in the clinical setting, the most interesting outcome is failure to rescue and this is a minor component of that outcome. I would like to better understand the implications of this research - for example, was the cost different for planned/unplanned or were the patient outcomes (morbidity/mortality) worse? That analysis could help to understand why this investigation is important.

Are there times where patients have to recover in the PACU because ICU bed space is not yet available? If so, how do you account for this because these patients would not actually be unplanned? If not, does that mean that your ICUs are underutilized?

The authors acknowledge the limitations of not having a testing and validation cohort. This does seem like a rather substantial limitation to any conclusions that could be drawn. There is not likely anything that can be done at this point, but it is a concern.

The authors also acknowledge the challenges of picking variables. I agree that this might be the biggest limitation to their analysis. I may not be understanding the analysis correctly, but wouldn't some of these variables be potential mediators for unplanned ICU admission. For example, PACU time is predictive of unplanned ICU admission, but wouldn't there be reasons that patients are spending more time in recovery and then to help clear recovery patients would be transferred to the ICU instead of the ward?

Did the authors investigate collinearity with the inclusion of so many variables?

Did the authors consider the potential use of a mixed model for healthcare providers (anesthesiologist or surgeon who did or did not plan ICU admission) to account for random effects?

It appears that the investigators used BMI as a continuous variable. The extremes are often the most interesting. I would be curious about a categorical variable as malnourished or underweight individuals (BMI<18) might behave differently. There are no other markers of malnutrition in their model.

With this data, it would be interesting to examine mortality after unplanned ICU admission. This description and better understanding of patients who experience failure to rescue would be even more clinically important.

The authors note the differences in their setting with high-resource areas like the United States, but it would be helpful for readers to also understand the context with low-resource settings. Given that the ICU bed space is even more limited in other areas of the world (https://doi.org/10.1016/S2214-109X(21)00291-6), the authors could comment on the utility of prediction in these settings particularly with interest in failure to rescue after surgery (DOI: 10.1097/sla.0000000000005215 and https://doi.org/10.1111/anae.14934).

Reviewer #3: In their interesting manuscript, the authors describe the course of patients with not anticipated secondary ICU admission in the period 2013-2017. They describe factors contributing to the need for ICU care and the outocme of this patient population. In general the manuscript is acceptable wqritten and require some language editing, the topic is of interest and clinical relevance for the community.

There are some topics which require attention and most likely a revised version of the manuscript.

1. The authors present data from 2013-2017, as we know the population of patients has been changing in the last years due to shared decision making and a change in surgical approach. Do the authors think that the population is still valid in 2022?

2. In table 4 the authors gave an overview about the underlying reason for ICU admission. I was wondering why sepis/sirs etc. was placed in the field cardiovascular= in particular as vasopleagia was placed in a different entity, please comment?

3. The authors used phenylehrine boluses and continuous vasopressor infusion as parameters, I think the use of vasopressors should be stated instead of phenylephrine.

4. The in-hospital mortality of the ICU group was 13.9%, it will be interesting if the patients died in the ICU or in the hospital, please add. (Table 2)

5. It is more common to describe it as Anesthesia technique instead of Method of Anesthesia, probably an exchange is good.

6. One of the patient populations at risk is the group of patients receiving betablockade for secondary prophylaxis. Is the use of an increased heart rate really justified. Do the authors used oher drugs than anti-coagulation drugs as part of the risk analysis and was the prescription behaviour consisten in the timeframe from 2013.-2017.

7. The group of general surgical patients contributed significantly two questions:

a. Are vascular surgical patients part of this group?

b. cany ou split up in cancer non-cancer surgery or do you think this is not neccessary?

8. For me cerebral infarction is rather cardiovascular then heamatological and major bleeding in my opinion does also not fit within that group, do you think it may be useful to re-group parts of this table? Have you thought about working with MACE like endpoints?

9. In the meantime, there are much better references for Bayesian network analysis than a textbook (probably some examples from the CoVid literature)

10. I found the number of patients with a saturation below 85% quite impressive, any references about this topic showing that this is in line with the benchmark in the literature?

11. The use of the word `more`seems not always appropriate, please do some language spelling,

12. The argumentation that data is used in a perido with an unchangend EMR is quite relevant.

12 For my interest, why do you use MATLAB AND SPSS programaming for the forward and backward analysis, is there reasoning?

Reviewer #4: In this study, the investigators used EMR data to predict unplanned ICU admissions. Prevalence of this outcome was rare, occurring in <1% of patients. This was a single center study with a fairly large sample size overall, although the authors did not provide how many patients were excluded due to missing covariates in the multiple regression model. The most important finding was that both peri- and pre-operative data contribute to improved prediction of ICU use. I have some concerns about the robustness of the performance evaluation presented. Also, justification for such a model as a decision support tool was lacking. Improving these areas of the paper would be necessary to justify publication. Specific comments are below.

• What decisions are affected by this model? It is described as a decision support tool, but no decisions are described. It is unclear when and how the model should be used.

• The STROBE diagram should include missing data and then the final boxes should be the dataset analyzed after excluding the missing patients

• AUC is not a robust measure for evaluating a prediction model when prevalence is very low – ICU admission rate in this study was <1%. Precision, recall, F1 should also be reported. Calibration and discrimination should also be evaluated.

• Treatment of the confounder described in lines 195-198 is vague and requires clarification

• Treatment of the continuous predictors as linear in the multivariable model should be justified. Did you first explore the relationships using cubic smoothing splines or lowess to determine if the relationships were approximately linear? ASA is more often treated as categorical, so justification for treating it as numerical/linear would be beneficial.

• Units of continuous predictors are small and result in very small odds ratios. I would consider presenting the odds ratios by 10 unit increases.

• Was collinearity explored? Presumably, surgery lowest HR and PACU lowest HR would be correlated. It would be preferable to present the most important predictors that don’t explain the same variability in Y.

6. PLOS authors have the option to publish the peer review history of their article (what does this mean?). If published, this will include your full peer review and any attached files.

Reviewer #1: **Yes: **Zakary Doherty

Reviewer #2: No

Reviewer #3: No

Reviewer #4: **Yes: **Kathryn Colborn

---

## [Author Response · Author response to Decision Letter 0]

30 Sep 2022

Dear reviewers, 

Thank you very very much for your time and effort to revise our manuscript and provide us with valuable suggestions to improve the manuscript. A full response can be found in the separate document Response to reviewers. 

Best regards, on behalf of the authors,

Eveline Mestrom

---

## [Decision Letter · Decision Letter 1]

16 Nov 2022

PONE-D-22-01611R1Prediction of postoperative patient deterioration and unanticipated intensive care unit admission using perioperative factorsPLOS ONE

Dear Dr. Mestrom,

Thank you for submitting your manuscript to PLOS ONE. After careful consideration, we feel that it has merit but does not fully meet PLOS ONE’s publication criteria as it currently stands. Therefore, we invite you to submit a revised version of the manuscript that addresses the points raised during the review process.

We look forward to receiving your revised manuscript.

Kind regards,

Sandro Pasquali, M.D., Ph.D.

Academic Editor

PLOS ONE

Additional Editor Comments:

Please carefully address all reviewers' comments. The manuscript needs to be improved along the lines of reviewers' comments, especially those from reviewer 4. Failing to meet reviewer requests will result in a rejection as methodological quality is needed to meet PlosOne requirements.

Reviewers' comments:

Reviewer's Responses to Questions

**Comments to the Author**

1. If the authors have adequately addressed your comments raised in a previous round of review and you feel that this manuscript is now acceptable for publication, you may indicate that here to bypass the “Comments to the Author” section, enter your conflict of interest statement in the “Confidential to Editor” section, and submit your "Accept" recommendation.

Reviewer #1: All comments have been addressed

Reviewer #2: (No Response)

Reviewer #4: (No Response)

2. Is the manuscript technically sound, and do the data support the conclusions?

Reviewer #1: Yes

Reviewer #2: Partly

Reviewer #4: No

3. Has the statistical analysis been performed appropriately and rigorously? 

Reviewer #1: Yes

Reviewer #2: I Don't Know

Reviewer #4: No

4. Have the authors made all data underlying the findings in their manuscript fully available?

Reviewer #1: Yes

Reviewer #2: Yes

Reviewer #4: No

5. Is the manuscript presented in an intelligible fashion and written in standard English?

Reviewer #1: Yes

Reviewer #2: Yes

Reviewer #4: Yes

6. Review Comments to the Author

Reviewer #1: I thank the reviewers for their changes and considering my comments, it is much appreciated.

Overall:

4 main problems (referring to section in my previous comments):

#1

Thank-you for the responses to my suggestions. It is now clear you are intending to build a model to establish risk of an unanticipated ICU admission that could be built into your EMR and automatically calculated and presented to clinicians. Thank you for only including the full model in the main paper, this makes the results much simpler to read.

#2

Thank-you for the response to my suggestions. This use of this variables is now much more clear.

#3

Thanky-you for the response to my suggestion. I recognise the limited number of patients with the rare outcome of unanticipated ICU admission which precludes using a large number of inclusion criterion. The use of the figure and median/IQR is very helpful and confirms that most admissions are occuring in the first few post-operative days. I

#4

Thank-you for the response to my suggestion. The intended use of the model is very reasonable and would definitely assist clinicians with deciding whether to refer their patient to ICU early in their stay to avoid unanticipated admissions post ward transfer.

Abstract:

Nil suggestions.

Introduction:

Nil suggestions.

Methods:

Line 124: suggest defining the HR acronym after heart rate so it is clear you are referring to a heart rate rule in Line 127.

Discussions:

Nil suggestions.

Results:

Nil suggestions.

Figures:

Nil suggestions.

Tables:

Nil suggestions.

Reviewer #2: Thank you for addressing the questions raised. Seemingly due to study design, there remain a number of issues that are limitations to the practical implementation of this model.

Reviewer #4: In this manuscript the authors present a multivariable logistic regression model for prediction of unplanned ICU admission after surgery. They use pre, peri, and post-operative data. This study was conducted in a relatively small data set from one institution. I have a number of concerns about the statistical methods, results, and bias in the data source that are outlined below. Because this is not novel work, some recent work in this area has not been cited, and the results are not generalizable, the significance is low.

• Data should be split into training and test sets, according to TRIPOD guidelines

• Model selection based on p-values (i.e., stepwise, forward, backward) is outdated, especially for rather small data samples. Penalized regression techniques are preferred for a large number of covariates in relation to the number of events. False discovery rate can be added as an additional technique for variable selection after penalized regression.

• Would the early post-operative data introduce confounding by indication? Wouldn’t confounding by indication be present when comparing ICU admission and mortality? At the very least, there appear to be a number of mediators in the data, and this is not addressed in the statistical approach.

• Data are referred to as singular, but they should be referred to as plural

• There would have been significant potential for multicollinearity of the predictors, but there is no mention of evaluation of this. Was this evaluated?

• Inspection of the transfusion of thrombocytes in the multivariable models is necessary. Why is the CI so large? Presumably this is because it was a very rare exposure, but it may also be highly correlated with another predictor. This may cause variance inflation.

• Other performance statistics should be shown. Examples include AUCPR, discrimination plot, calibration plot. This is especially important for rare events, because AUC will be inflated.

• This small sample of patients and issues with bias in the data makes the results ungeneralizable.

• More ideas around how this model could be implemented in practice would strengthen the discussion

• Inclusion of ROC curves does not add value to the paper.

• Many of the labels for variables in the S3 table do not match the labels in Table 1, so it’s difficult to compare univariable and multivariable ORs.

• Some recent papers published on predictors of ICU admission after surgery are missing.

7. PLOS authors have the option to publish the peer review history of their article (what does this mean?). If published, this will include your full peer review and any attached files.

Reviewer #1: **Yes: **Zakary Doherty

Reviewer #2: No

Reviewer #4: No

---

## [Author Response · Author response to Decision Letter 1]

30 Jan 2023

Response to reviewers

Reviewer #1: I thank the reviewers for their changes and considering my comments, it is much appreciated.

Overall:

4 main problems (referring to section in my previous comments):

#1

Thank-you for the responses to my suggestions. It is now clear you are intending to build a model to establish risk of an unanticipated ICU admission that could be built into your EMR and automatically calculated and presented to clinicians. Thank you for only including the full model in the main paper, this makes the results much simpler to read.

#2

Thank-you for the response to my suggestions. This use of this variables is now much more clear.

#3

Thanky-you for the response to my suggestion. I recognise the limited number of patients with the rare outcome of unanticipated ICU admission which precludes using a large number of inclusion criterion. The use of the figure and median/IQR is very helpful and confirms that most admissions are occuring in the first few post-operative days. I

#4

Thank-you for the response to my suggestion. The intended use of the model is very reasonable and would definitely assist clinicians with deciding whether to refer their patient to ICU early in their stay to avoid unanticipated admissions post ward transfer.

Abstract:

Nil suggestions.

Introduction:

Nil suggestions.

Author response: Thank you very much for this positive response!

Methods:

Line 124: suggest defining the HR acronym after heart rate so it is clear you are referring to a heart rate rule in Line 127.

Author response: Thank you, that is a good idea. 

Discussions:

Nil suggestions.

Results:

Nil suggestions.

Figures:

Nil suggestions.

Tables:

Nil suggestions.

Reviewer #2: Thank you for addressing the questions raised. Seemingly due to study design, there remain a number of issues that are limitations to the practical implementation of this model.

Author response: Thank you for your time. We hope the limitations were addressed according to your expectations. 

Reviewer #4: In this manuscript the authors present a multivariable logistic regression model for prediction of unplanned ICU admission after surgery. They use pre, peri, and post-operative data. This study was conducted in a relatively small data set from one institution. I have a number of concerns about the statistical methods, results, and bias in the data source that are outlined below. Because this is not novel work, some recent work in this area has not been cited, and the results are not generalizable, the significance is low.

• Data should be split into training and test sets, according to TRIPOD guidelines

Author response: The reviewer points out one of the challenges in this study. We would like to thank the reviewer for the opportunity to improve our manuscript by implementing stratified bootstrapping to create, train and test the data. Instead, we implemented stratified bootstrapping to create train, test splits. In this way, we keep the size of the training dataset the same as the original and the number of patients in the outcome group stays the same. 

• Model selection based on p-values (i.e., stepwise, forward, backward) is outdated, especially for rather small data samples. Penalized regression techniques are preferred for a large number of covariates in relation to the number of events. False discovery rate can be added as an additional technique for variable selection after penalized regression.

Author response: Thank you for suggesting to perform penalized regression analysis. The results were added to the manuscript. In short, the penalized regression showed comparable predictors, AUROC and AUPRC compared to our previous analysis. 

The sections Methods, Results and Discussion were modified according to this new analysis. 

• Would the early post-operative data introduce confounding by indication? 

Author response: the early postoperative date includes only data from the PACU period. This implies that confounding by indication should be minimized as that sort of confounding is mostly induced when the patient is in the ward. 

Wouldn’t confounding by indication be present when comparing ICU admission and mortality? At the very least, there appear to be a number of mediators in the data, and this is not addressed in the statistical approach.

Author response: The reviewer points out the duality of presenting mortality data. In the first revision round, it was requested to insert mortality data.

This reviewer’s suggestion was taken into account and the mortality rates were added to the Result’s section in Line 184. In such a small group, it will be hard to draw valid conclusions for subgroup of mortality after unanticipated ICU admission. 

The ultimate goal would be to decrease morbidity and mortality but the study is not powered to provide an answer. 

• Data are referred to as singular, but they should be referred to as plural

Author response: Thank you for notifying this inconsistency. We checked every sentence with ‘data’ and changed singular into plural throughout the manuscript. This adjusting applied to Line 172 and 245. 

• There would have been significant potential for multicollinearity of the predictors, but there is no mention of evaluation of this. Was this evaluated?

Author response: We agree with the reviewer that multicollinearity might play a role between the different predictors. We did not report on the evaluation of multicollinearity because it is important to consider when studying causality and because multicollinearity influences p values and confidence intervals but does not influence prediction. The evaluation was performed in SPSS using correlation matrix, which showed correlation values between 0.001 to 0.334 with the majority below 0.100, suggesting low correlations. The magnitude of the standard error for each variable was inspected and within acceptable ranges. If the reviewer prefers, the results of these analysis can be added as supplemental material. 

• Inspection of the transfusion of thrombocytes in the multivariable models is necessary. Why is the CI so large? Presumably this is because it was a very rare exposure, but it may also be highly correlated with another predictor. This may cause variance inflation.

Author response: The reviewer is right in the assumption that the transfusion of thrombocytes was very rare. In the multicollinearity analysis, there was no correlation with, for example, erythrocytes, suggesting major bleeding complication. After the implementation of penalized regression analysis instead of forward-backward selection analysis, the thrombocytes were no longer important for prediction. 

• Other performance statistics should be shown. Examples include AUCPR, discrimination plot, calibration plot. This is especially important for rare events, because AUC will be inflated.

Author response: The AUPRC plot was added to the manuscript as Fig 4. according to the reviewer’s relevant suggestion.

• This small sample of patients and issues with bias in the data makes the results ungeneralizable.

Author response: We agree with the reviewer that the small sample is a limitation. However, the analyzed dataset is larger than comparable studies and can still:

1) serve as an example of reproducibility of previous studies

2) provide base for local prediction models

3) show the importance of including data from different phases in and around surgery that is readily available and easily found in the EMR

• More ideas around how this model could be implemented in practice would strengthen the discussion

Author response: In the previous revision round, the manuscript was modified to provide more ideas about implementation in clinical practice. Based on this reviewer’s point, we added the following to the Discussion section in Line 310-312: “This digital tool could automatically calculate a low, intermediate, or high risk of unanticipated ICU admission and provide decision support to the anesthesiologist in PACU.” 

• Inclusion of ROC curves does not add value to the paper.

Author response: In medical journals, the ROC curve is frequently used, requested and interpreted, even though it might not be perfect from a statistical point of view. Therefore, we prefer to leave it in the manuscript. In addition, in the previous revision round, the extra two ROC curves were removed from the manuscript and includes as supplemental material. 

• Many of the labels for variables in the S3 table do not match the labels in Table 1, so it’s difficult to compare univariable and multivariable ORs.

Author response: The labels were matched more clearly. 

• Some recent papers published on predictors of ICU admission after surgery are missing.

Author response: Most papers examine factors from a different point of view, namely once the patient has stayed on the ward. Possibly, the manuscript is not clear enough in the perspective of this study where we were searching for prediction of unanticipated ICU admission from the point of view of an anesthesiologist in post anesthesia care unit. A literature search was performed once more but did not result in recent papers from this PACU perspective. We are wondering if the reviewer could mention the papers the reviewer had in mind?

---

## [Decision Letter · Decision Letter 2]

14 Mar 2023

PONE-D-22-01611R2Prediction of postoperative patient deterioration and unanticipated intensive care unit admission using perioperative factorsPLOS ONE

Dear Dr. Mestrom,

Thank you for submitting your manuscript to PLOS ONE. After careful consideration, we feel that it has merit but does not fully meet PLOS ONE’s publication criteria as it currently stands. Therefore, we invite you to submit a revised version of the manuscript that addresses the points raised during the review process.

We look forward to receiving your revised manuscript.

Kind regards,

Sandro Pasquali, M.D., Ph.D.

Academic Editor

PLOS ONE

Additional Editor Comments:

Please carefully address comments from reviewer 2, discussing limitation of this study in the Discussion section. Please, also address comments from reviewer 5.

Reviewers' comments:

Reviewer's Responses to Questions

**Comments to the Author**

1. If the authors have adequately addressed your comments raised in a previous round of review and you feel that this manuscript is now acceptable for publication, you may indicate that here to bypass the “Comments to the Author” section, enter your conflict of interest statement in the “Confidential to Editor” section, and submit your "Accept" recommendation.

Reviewer #1: All comments have been addressed

Reviewer #2: (No Response)

Reviewer #5: (No Response)

2. Is the manuscript technically sound, and do the data support the conclusions?

Reviewer #1: (No Response)

Reviewer #2: Partly

Reviewer #5: Partly

3. Has the statistical analysis been performed appropriately and rigorously? 

Reviewer #1: (No Response)

Reviewer #2: I Don't Know

Reviewer #5: Yes

4. Have the authors made all data underlying the findings in their manuscript fully available?

Reviewer #1: (No Response)

Reviewer #2: Yes

Reviewer #5: No

5. Is the manuscript presented in an intelligible fashion and written in standard English?

Reviewer #1: (No Response)

Reviewer #2: Yes

Reviewer #5: Yes

6. Review Comments to the Author

Reviewer #1: (No Response)

Reviewer #2: As before, the study design limits the practical implementation of this model.

...............................................................................................................................................

Reviewer #5: I thank the authors for the opportunity of reviewing this interesting manuscript. The topic is important and the study was well performed. The authors have addressed the requests by previous reviewers.

The message of the authors is clear but I have some doubts about the translation from theory to practice of the predictive model developed.

These are my suggestions for the authors:

The authors analyzed and built the model on data collected between 2013 and 2017. Some changes in daily practice may have occurred since 2017. This may limit the application of the predictive model. Authors should explain and comment on this.

I am not confident with penalized logistic regression which is not commonly used. This probably helps to better assess the readability of the manuscript for ordinary readers... I am puzzled by some problems:

(1) Why did the authors not report P-values for each variable in Table 1?

(2) Table 1 shows the "Most important predictors". I guess the authors mean that these are the significant predictors that are part of the final model. I suggest correcting the table title accordingly. In addition, the authors should also report insignificant predictors (perhaps as supplementary material). For example: what about ASA class 2?

(3) Table 1: Increased BMI appears to be protective... this deserves a comment in the discussion section

(4) The final model is poorly reported. How should it work in daily practice?

(5) I do not understand what the authors mean with “Lowest heart rate” and “Highest heart rate”

7. PLOS authors have the option to publish the peer review history of their article (what does this mean?). If published, this will include your full peer review and any attached files.

Reviewer #1: **Yes: **Zakary Doherty

Reviewer #2: No

Reviewer #5: No

---

## [Author Response · Author response to Decision Letter 2]

28 Apr 2023

Dear editor and reviewers, 

In the 'Attach files' section, the revised documents and a response to reviewers questions was uploaded. Even though we took the time to answer until the deadlines, we hope you will appreciate the manuscript for publication now. 

Best regards,

Eveline Mestrom

---

## [Decision Letter · Decision Letter 3]

24 May 2023

Prediction of postoperative patient deterioration and unanticipated intensive care unit admission using perioperative factors

PONE-D-22-01611R3

Dear Dr. Mestrom,

We’re pleased to inform you that your manuscript has been judged scientifically suitable for publication and will be formally accepted for publication once it meets all outstanding technical requirements.

Kind regards,

Sandro Pasquali, M.D., Ph.D.

Academic Editor

PLOS ONE

Additional Editor Comments (optional):

The Authors addressed reviewersì comments.

Reviewers' comments:

Reviewer's Responses to Questions

**Comments to the Author**

1. If the authors have adequately addressed your comments raised in a previous round of review and you feel that this manuscript is now acceptable for publication, you may indicate that here to bypass the “Comments to the Author” section, enter your conflict of interest statement in the “Confidential to Editor” section, and submit your "Accept" recommendation.

Reviewer #5: All comments have been addressed

2. Is the manuscript technically sound, and do the data support the conclusions?

Reviewer #5: Yes

3. Has the statistical analysis been performed appropriately and rigorously? 

Reviewer #5: Yes

4. Have the authors made all data underlying the findings in their manuscript fully available?

Reviewer #5: No

5. Is the manuscript presented in an intelligible fashion and written in standard English?

Reviewer #5: Yes

6. Review Comments to the Author

Reviewer #5: The authors reviewed the manuscript as required. I have no further suggestions to make. I congratulate them for the interesting work done.

7. PLOS authors have the option to publish the peer review history of their article (what does this mean?). If published, this will include your full peer review and any attached files.

Reviewer #5: No

---

## [Editor Report · Acceptance letter]

26 Jul 2023

PONE-D-22-01611R3 

Prediction of postoperative patient deterioration and unanticipated intensive care unit admission using perioperative factors 

Dear Dr. Mestrom:

I'm pleased to inform you that your manuscript has been deemed suitable for publication in PLOS ONE. Congratulations! Your manuscript is now with our production department. 

Kind regards, 

on behalf of

Dr. Sandro Pasquali 

Academic Editor

PLOS ONE